# Nestin^+^ Mesenchymal Precursors Generate Distinct Spleen Stromal Cell Subsets and Have Immunomodulatory Function

**DOI:** 10.3390/ijms231911819

**Published:** 2022-10-05

**Authors:** Jing Huang, Ronghai Deng, Weiqiang Li, Meihua Jiang, Andy Peng Xiang, Xiaoran Zhang

**Affiliations:** 1Center for Stem Cell Biology and Tissue Engineering, Key Laboratory for Stem Cells and Tissue Engineering, Ministry of Education, Sun Yat-sen University, Guangzhou 510080, China; 2Department of Organ Transplantation, The First Affiliated Hospital of Sun Yat-sen University, Guangzhou 510080, China; 3Department of Biochemistry, Zhongshan School of Medicine, Sun Yat-sen University, Guangzhou 510080, China; 4Guangzhou Regenerative Medicine and Health Guangdong Laboratory, Guangzhou 510080, China

**Keywords:** mesenchymal stromal cells, spleen, Nestin, immunoregulation, inflammatory bowel diseases

## Abstract

Mesenchymal stromal cells (MSCs) are known to be widespread in many tissues and possess a broad spectrum of immunoregulatory properties. They have been used in the treatment of a variety of inflammatory diseases; however, the therapeutic effects are still inconsistent owing to their heterogeneity. Spleen stromal cells have evolved to regulate the immune response at many levels as they are bathed in a complex inflammatory milieu during infection. Therefore, it is unknown whether they have stronger immunomodulatory effects than their counterparts derived from other tissues. Here, using a transgenic mouse model expressing GFP driven by the Nestin (*Nes*) promoter, *Nes*-GFP^+^ cells from bone marrow and spleen were collected. Artificial lymphoid reconstruction in vivo was performed. Cell phenotype, inhibition of T cell inflammatory cytokines, and in vivo therapeutic effects were assessed. We observed *Nes*-GFP^+^ cells colocalized with splenic stromal cells and further demonstrated that these *Nes*-GFP^+^ cells had the ability to establish ectopic lymphoid-like structures in vivo. Moreover, we showed that the *Nes*-GFP^+^ cells possessed the characteristics of MSCs. Spleen-derived *Nes*-GFP^+^ cells exhibited greater immunomodulatory ability in vitro and more remarkable therapeutic efficacy in inflammatory diseases, especially inflammatory bowel disease (IBD) than bone marrow-derived *Nes*-GFP^+^ cells. Overall, our data showed that *Nes*-GFP^+^ cells contributed to subsets of spleen stromal populations and possessed the biological characteristics of MSCs with a stronger immunoregulatory function and therapeutic potential than bone marrow-derived *Nes*-GFP^+^ cells.

## 1. Introduction

Mesenchymal stromal cells (MSCs) were originally considered nonhematopoietic stem cells isolated from bone marrow that participate in the formation of the bone marrow hematopoietic microenvironment and play an obvious supporting role in the proliferation and differentiation of hematopoietic stem cells [1]. They can self-renew, differentiate into osteoblasts, adipocytes and chondrocytes, and have a fibroblast-like morphology with notable plastic-adherent properties. In recent years, scientists have explored the intrinsic physiological roles that MSCs play in vivo and identified nutritional support and immune regulation functions, which highlight their therapeutic potential in tissue injury repair and inflammatory disease. For instance, the antimicrobial features of MSCs are linked to the paracrine release of several antimicrobial peptides and immunomodulatory factors, and the MSC treatment is a promising antibiotic-free therapy strategy for bacterial diseases [2].

Exogenous MSC-related clinical applications involve graft versus host disease [3], osteoarthritis [4], respiratory diseases [5], rheumatoid arthritis [6], lupus [7], myocardial infarct [8], stroke [9], and inflammatory bowel disease [10]. Although many preclinical experiments choose bone marrow-derived MSCs for cell-based therapies, limited evidence has proven that MSCs from different sources differ in tissue repair and immunomodulatory ability, suggesting that we might look for better sources for specific diseases.

Large numbers of studies have proven that the main mechanisms of MSC therapy are the regulation of the development, differentiation and function of B and T lymphocytes [11]. The spleen is the main site for the development of T and B lymphocytes in adult tissues [12]. Stromal cell subpopulations such as follicular dendritic cells (FDCs) and marginal reticular cells (MRCs) in the B lymphocyte region and fibroblast reticular cells (FRCs) in the T lymphocyte region play an important role in the survival, differentiation and maturation of lymphocytes in the spleen [13]. Previous studies have found that FRCs, MRCs and FDCs express several cell surface markers similar to MSCs, such as CD73, CD90, CD105, CD157, PDGFRa, and Sca-1 [14], suggesting that stromal cells in the spleen may have a functional correlation with MSCs. Karin Tarte et al. found that bone marrow-derived MSCs showed phenotypes of FRCs under the stimulation of TNF-α and lymphotoxin-α1β2 [15]. Tonsil-derived MSCs exhibit the characteristics of FDCs stimulated by cytokines and TLR3 agonists. Moreover, FRCs have strong immunosuppressive abilities, including inducing immune tolerance, inhibiting T cell proliferation, and promoting Treg proliferation to prevent the occurrence of autoimmune diseases, which is similar to the immunomodulatory function of MSCs [16]. Similarly, Fletcher al et al. showed that FRCs isolated from lymph nodes were far more effective than bone marrow-derived MSCs in the treatment of septic mice [17]. Therefore, we wanted to explore whether stromal cells residing in the spleen can be defined as spleen-derived MSCs and whether a complex immune environment might endow them with stronger immunomodulatory ability and better clinical application potential than bone marrow-derived MSCs. However, the lack of a common marker to identify MSCs from different tissues poses a limitation when comparing the general performance of different isolated populations. The phenotype and markers of mesenchymal stem/progenitor cells in adult spleens remain unknown.

Nestin, originally found in neuroepithelial stem cells, is an intermediate filament protein expressed in the early stages of development [18]. More importantly, Nestin is also expressed in some adult stem/progenitor cell populations, indicating that Nestin might be a common marker of multipotent stem cells. Using Nestin as a tracking marker, various adult stem cells have been prospectively isolated, including mesenchymal stem cells and hair follicle stem cells, suggesting that it can be used as a specific marker for isolating tissue-resident MSCs [19,20,21,22].

Here, we report a method for the identification and isolation of stromal cells in the spleen on the basis of Nestin expression using flow cytometry in combination with in vitro functional assays. Our assays provided evidence that these cell lines exhibited MSC characteristics and further demonstrated that Nestin-positive stromal cells in the spleen had the capacity to partially restore lymph node construction in vivo following transplantation into the kidney capsule and possessed a greater immunoregulatory function than bone marrow-derived MSCs.

## 2. Results

### 2.1. Characterization of Nestin Expression in the Spleen

To identify the expression of Nestin in the mouse spleen, we assessed the GFP signal in the *Nes*-GFP transgenic mouse model expressing GFP driven by the Nestin (*Nes*) promoter using confocal microscopy. Similar to findings in the testis, kidney and heart [20,21,23], we also observed GFP expression in the mouse spleen (Figure 1A). Additionally, we confirmed the expression of Nestin in the mouse spleen using *Nes*-Cre mice bred with Rosa26-td-Tomato (R26RtdT) reporter mice (*Nes*-Cre; R26RtdT) (Appendix A). To systematically evaluate the change in Nestin expression during development, we performed quantitative RT-PCR (qRT-PCR) to detect the mRNA expression levels of Nestin in the whole mouse spleen at 7, 14, 28, and 90 days of age. Our result showed a significant peak in Nestin expression on day 14 (Figure 1B), which gradually decreased on days 28 and 90 after birth.

To further characterize *Nes*-GFP^+^ cells and their correlation with stromal cells in the spleen, we examined the expression of several typical splenic stromal cell lineage markers, ER-TR7, CD35, and MadCAM-1, representing fibroblastic reticular cells (FRCs), follicular dendritic cells (FDCs), and marginal reticular cells (MRCs), respectively [24,25,26]. Immunofluorescence analysis of adult spleens from *Nes*-GFP mice revealed the GFP^+^ cells colocalized with ER-TR7, CD35, and MadCAM-1 in the white pulp (WP) (Figure 1C–E). Assessment of ER-TR7, CD35, and MadCAM-1 expression and quantification of GFP^+^ cells showed that the large majority of FRCs, FDCs, MRCs were GFP positive, thus indicating a contribution of Nestin^+^ cells to WP zone stromal cells (Figure 1F). We also detected the expression of other regular stromal cell markers: ICAM1, VCAM1, and CD31. Similarly, Nestin colocalized with ICAM1 and VCAM1, which are markers of lymphoid tissue organizer (LTo) cells [27] (Appendix A), instead of CD31 (Appendix A). Taken together, these results indicate that *Nes*-GFP^+^ cells in the mouse spleen may be an important subpopulation of spleen stromal cells.

### 2.2. Isolation and Characterization of Nes-GFP^+^ Cells from the Spleen

We sorted cells from the spleens of *Nes*-GFP transgenic mice using flow cytometric sorting and found that *Nes*-GFP^+^ cells constituted about 0.68% of the total spleen cell population (Figure 2A,B). As Nestin has been identified as a potential marker for MSCs in various tissues [20,21,23,28], we wondered whether *Nes*-GFP^+^ cells in the spleen could be splenic MSCs. To confirm this hypothesis, we first carried out a single-cell sphere formation assay in which single-cell suspensions derived from primary *Nes*-GFP^+^ cells were seeded by serial dilution into 96-well plates to demonstrate the self-renewal capacity of the *Nes*-GFP^+^ cells. After a 10-day culture in vitro, the isolated spleen-derived *Nes*-GFP^+^ cells yielded many clonal spheres demonstrating a strong proliferative capacity in vitro (Figure 2C). We also performed CCK8 to confirm the proliferation ability of *Nes*-GFP^−^ cells and *Nes*-GFP^+^ cells from the spleen, and our results showed that *Nes*-GFP^−^ cells barely had the proliferation ability and it was difficult to achieve the in vitro expansion (Appendix A). Moreover, we performed in vitro cytokine assays to detect immunosuppressing effects and did not observe significant immunosuppressing effects in *Nes*-GFP^−^ cells (Appendix A). Second, we cultured these cells under defined differentiation conditions to characterize the trilineage differentiation capacity. We observed that the *Nes*-GFP^+^ cells could differentiate into osteocytes, adipocytes, and chondrocytes, as identified by Alizarin red, Oil Red O, and toluidine blue staining, respectively (Figure 2D). Third, we used flow cytometry to analyze some MSC-related surface markers in *Nes*-GFP^+^ cells. As expected, these cells were positive for CD29, CD90, CD106 and PDPN, but expressed 0.05% for CD11c, 8.29% for CD31, and 6.48% for CD45 (Figure 2E). Compared with *Nes*-GFP^+^CD31^-^CD45^-^ cells, *Nes*-GFP^+^CD31^+^ and *Nes*-GFP^+^CD45^+^ cells scarcely expressed spleen stromal cell markers, such as CXCL13, CCL19, CCL21 (Appendix A), and *Nes*-GFP^+^ CD31^+^ and *Nes*-GFP^+^ CD45^+^ cells also expressed low levels of PDPN (Appendix A). Given the MSC-like stem cell features of *Nes*-GFP^+^ cells isolated from spleens, we proposed that Nestin may be a potential marker of spleen-derived MSCs.

### 2.3. Spleen-Derived Nes-GFP^+^ Cells Could Form Secondary Lymphoid Organs

Our results suggest that *Nes*-GFP^+^ cells could be the stem/progenitor cells of lymphoid stromal cells in the spleen, so we hypothesized that spleen-derived *Nes*-GFP^+^ cells have the potential to remain in the stromal network of lymphoid organs. To investigate this hypothesis, we performed artificial lymphoid formation [29] to detect the ability of *Nes*-GFP^+^ cells to form intact lymphoid tissue. After isolating CD45^-^*Nes*-GFP^-^ and CD45^-^*Nes*-GFP^+^ cells from *Nes*-GFP transgenic mice, we transplanted these cells with collagen scaffolds into the kidney capsule of C57BL/6 wild-type mice for three weeks, and then stained the separated transplants with antibodies against T cell marker (CD3) and B cell marker (B220). The results of immunofluorescence staining showed a clear T cell zone and B cell zone in the transplants formed by CD45^-^Nestin^+^ cells (Figure 3A). Compared with the transplant formed by spleen-derived *Nes*-GFP^-^ cells, the transplant formed by spleen-derived *Nes*-GFP^+^ cells had a similar structure to the lymph node, which suggested that spleen-derived *Nes*-GFP^+^ cells could form intact lymphoid tissue-like organoids (Figure 3B–D). Taken together, our results indicate that *Nes*-GFP^+^ cells derived from the spleen have the ability to rebuild the lymphatic stromal cell network.

### 2.4. Immunosuppressive Properties of Nes-GFP^+^ Cells Derived from the Spleen

Bone marrow-derived MSCs are known to suppress the function of T cells. Our results showed that spleen-derived *Nes*-GFP^+^ cells exhibit similar properties to MSCs, so we hypothesized that spleen-derived *Nes*-GFP^+^ cells could also have immunosuppressive properties. To investigate the immunoregulatory abilities of *Nes*-GFP^+^ cells, we detected their effects on the proinflammatory cytokine production of CD3^+^ T cells. After coculture with CD3^+^ T cells for 48 h, the flow cytometry results showed that spleen-derived *Nes*-GFP^+^ cells significantly reduced the secretion of TNF-α (Figure 4A,B) and IFN-γ (Figure 4C,D) by CD3^+^ T cells, and could even remain the effects at a ratio of 1:20. Moreover, compared with that of bone marrow-derived *Nes*-GFP^+^ cells, spleen-derived *Nes*-GFP^+^ cells showed an even stronger suppressive effect at the same ratio (Figure 4E). Thereinto, spleen-derived *Nes*-GFP^+^ cells could also markedly decreased the secretion of CD3^+^CD8^+^TNF-α^+^ T cells (Appendix A) and CD3^+^CD8^+^IFN-γ^+^ T cells (Appendix A), while bone marrow-derived *Nes*-GFP^+^ cells had no inhibition effects at a ratio of 1:20 on CD3^+^CD8^+^TNF-α^+^ or CD3^+^CD8^+^IFN-γ^+^ T cells (Appendix A–D). Furthermore, T cell proliferation and activation experiments showed that both bone marrow and spleen-derived *Nes*-GFP^+^ cells had observably inhibitory effects on T cell proliferation and activation, manifested as down-regulated expressions of activated marker CD69 and CD25, and the inhibitory effects of spleen-derived *Nes*-GFP^+^ cells were more obvious (Figure 5A–H). Overall, the results revealed that spleen-derived *Nes*-GFP^+^ cells exhibit stronger immunosuppressive properties than bone marrow-derived *Nes*-GFP^+^ cells in vitro.

### 2.5. Spleen-Derived Nes-GFP^+^ Cells as a Potential Source for the Treatment of Inflammatory Diseases

MSCs have shown significant therapeutic potential in many clinical trials of inflammatory diseases, so we postulated that spleen-derived *Nes*-GFP^+^ cells could also attenuate inflammation in vivo. To evaluate the treatment prospects of spleen-derived *Nes*-GFP^+^ cells, we compared the therapeutic potential of bone marrow-derived and spleen-derived *Nes*-GFP^+^ cells in TNBS-induced colitis. Our results showed that both of them alleviated TNBS-induced colitis by injection intravenously, but spleen-derived *Nes*-GFP^+^ cells exhibited a better therapeutic efficacy (Figure 6A–F). Specifically, the group treated with spleen-derived *Nes*-GFP^+^ cells showed reduced body weight loss and lower DAI scores (Figure 6A,B). Additionally, the average colon length was reduced to a lesser degree in the spleen-derived *Nes*-GFP^+^ cells group than the bone marrow-derived *Nes*-GFP^+^ cells group (Figure 6C,D). Moreover, the histopathologic analysis indicated that epithelium loss and inflammatory cell infiltration were more significantly reduced after being treated by spleen-derived *Nes*-GFP^+^ cells (Figure 6E,F).

We also compared the effects of spleen-derived and bone marrow-derived *Nes*-GFP^+^ cells on contact hypersensitivity (CHS) mice. We injected spleen-derived and bone marrow-derived *Nes*-GFP^+^ cells separately into CHS mice and measured ear swelling and cellular infiltration (Appendix A). Histopathological examination revealed that ear swelling and cellular infiltration were improved after the injection of *Nes*-GFP^+^ cells (Appendix A). Additionally, analysis of ear thickness after spleen-derived *Nes*-GFP^+^ cell injection showed a peak at 48 h and then a gradual decrease. A similar trend was observed in bone marrow-derived *Nes*-GFP^+^ cell-injected mice, but ear swelling was more severe than that in spleen-derived *Nes*-GFP^+^ cell-injected mice (Appendix A). Moreover, *Nes*-GFP^+^ cells derived from the spleen showed a better potential to reduce proinflammatory cytokine-producing CD8^+^ effector T cells in vivo (Appendix A). Taken together, our results suggest that *Nes*-GFP^+^ cells isolated from the spleen have better immunomodulatory function than *Nes*-GFP^+^ cells isolated from bone marrow in vivo.

## 3. Discussion

In this study, we sought to examine whether Nestin can serve as a novel marker for the identification and isolation of spleen stromal cells. We isolated *Ne*s-GFP^+^ cells from the spleens of *Nes*-GFP transgenic mice and found that these cells had characteristics consistent with splenic mesenchymal cell subset population precursors. *Nes*-GFP^+^ cells in the spleen not only proliferated continuously but also showed highly clonogenic potential, which suggests that *Nes*-positive stromal cells were capable of self-renewal. Moreover, the cells showed mesenchymal lineage differentiation potential. In addition, *Nes*-GFP^+^ stromal cells derived from the spleen showed greater immunomodulation ability than bone marrow-derived *Nes*-GFP^+^ cells, as evidenced by the suppression of the proliferation and functions of T lymphocytes in vitro and the improvement observed upon intravenous injection into IBD and CHS mice.

The intermediate filament protein Nestin is a widely employed marker of multipotent neural stem cells (NSCs) and we have previously shown that Nestin is important for the survival and self-renewal of NSCs during embryonic development [30]. In adults, Nestin expression is frequently detected in areas of regeneration, such as neural progenitors, mesenchymal stem cells, hair follicle stem cells and myogenic cells, where Nestin-positive cells might function as a reservoir of stem/progenitor cells capable of proliferation and differentiation. Therefore, we proposed Nestin as a candidate marker for splenic stromal cells, including FRCs, FDCs and MRCs. Moreover, *Nes*-GFP^+^ cells derived from the spleen also encompass a subset of stromal progenitor cells with lymphoid tissue organizer activity capable of supporting ectopic lymphoid-like structures under the kidney capsule. Previous studies by Castagnaro and colleagues confirmed that Nkx2-5^+^/Islet1^+^ mesenchymal precursors generate distinct spleen stromal cell subsets [31]. Nike Julia Krautler et al. also proved that FDCs arise from ubiquitous perivascular precursors expressing PDGFR-β [32]. Based on our work, we identified Nestin as a potential marker of a distinct subset of splenic stromal cells, and *Nes*-positive cells might act as splenic stromal progenitor cells. However, we also identified a small proportion that expressed CD31 and CD45, which suggested that Nestin might mark cells that belong to the mesenchymal stromal as well as the endothelial subsets or lymphoid subsets.

As the largest peripheral lymphoid organ, splenic stromal cells orchestrate highly structured microenvironments that maximize the efficacy of immune responses to pathogens. In the T cell area, FRCs associate with clusters of T cells and support the formation of a conduit network that facilitates the directed migration of hematopoietic cells [33]. In the B cell area, FDCs capture and present antigens to B cells and support the process of affinity maturation; in the marginal zone area, marginal reticular cells provide structural support and participate in conduit formation [26,34]. Based on a complex immune milieu, it was deduced that splenic stromal cells might be endowed with higher efficacy in immunoregulation than mesenchymal stromal cells from other tissue sources. Here, we demonstrate that when expanded ex vivo and cocultured with T cells, spleen Nestin^+^ MSCs showed greater immunocompetence than bone marrow Nestin^+^ MSCs in the control of T cell inflammatory factor secretion. When used in mouse inflammatory disease therapy, such as IBD and CHS, spleen Nestin^+^ MSCs significantly controlled the inflammatory response, and the therapeutic effect was better than bone marrow Nestin ^+^ MSCs. Additionally, we previously reported that MSCs could significantly promote the production of regulatory B cells and significantly regulate the differentiation direction of B cells [35,36]. With special localization of spleen Nestin^+^ MSCs, it is speculated they may have regulation on the number and function of B cells and deserves further study.

In conclusion, we identified Nestin as a potential marker of splenic stromal cells, and splenic *Nes*-GFP^+^ cells exhibit the characteristics of mesenchymal stromal cells. Moreover, transplanted splenic *Nes*-GFP^+^ cells under the kidney capsule appeared to organize the formation of lymphoid compartments, which reminded us of LTo cells. Additionally, an in vivo IBD and CHS disease model and in vitro coculture experiments demonstrated that *Nes*-GFP^+^ cells derived from the spleen were excellent in immunomodulation. Thus, our findings provide new insight into an optimal source of mesenchymal stromal cells for cell-based therapy in overwhelming inflammatory and autoimmune diseases and inspire the generation of artificial secondary lymphoid organs.

## 4. Materials and Methods

### 4.1. Animals

Homozygous Nestin (*Nes*)-GFP transgenic mice on the C57BL/6 genetic background were provided by Dr Masahiro Yamaguchi [37]. C57BL/6 wild-type mice were purchased from the Animal Center at the Medical Laboratory of Guangdong Province, China. *Nestin-cre* mice (Stock No: 003771) and *Rosa26-td-Tomato* (*R26RtdT*) reporter mice (Stock No: 007905) were obtained from the Jackson Laboratory. All specific pathogen-free mice used for studies were housed and handled according to protocols approved by the Sun Yat-sen University Institutional Animal Care and Use Committee.

### 4.2. Isolation and Culture of Nes-GFP^+^ Cells from Nes-GFP Transgenic Mice

*Nes*-GFP^+^ cells derived from mouse spleens were collected as previously described [20]. Briefly, the spleens were minced into small pieces with 5 mL of HBSS digestive solution including Type IV collagenase and incubated in a water bath at 37 °C for 30 min. After adding DMEM containing 10% fetal bovine serum (FBS, HyClone, Logan, UT, USA) to stop digestion, the cell suspensions were centrifuged at 1250× *g* for 5 min at room temperature, washed twice with phosphate-buffered saline (PBS) and filtered through a 40-μm cell strainer to prepare a single-cell suspension. *Nes*-GFP^-^ cells and *Nes*-GFP^+^ cells were sorted using flow cytometry (Influx, BD) and cultured in DMEM/F12 (1:1) containing 1 ng/mL LIF (Millipore, Burlington, MA, USA), 5 μg/L insulin-transferrin-sodium selenite (ITS, Sigma-Aldrich, St. Louis, MI, USA), 5% chicken embryo extract (US Biologicals, Salem, MA, USA), 0.1 mM β-mercaptoethanol (Invitrogen, Waltham, MA, USA), 1% nonessential amino acids (HyClone), 1% N2 (Invitrogen), 2% B27 (Invitrogen), 20 ng/mL bFGF (Invitrogen), and 20 ng/mL EGF (PeproTech, London, UK). These cells were kept at 37 °C in a humidified 5% CO_2_ water-jacketed incubator and were propagated every 3 days. *Nes*-GFP^+^ cells from similar passages were used in all assays.

### 4.3. RNA Isolation and Quantitative Real-Time and Reverse Transcription PCR

Total RNA was extracted from spleen tissues using TRIzol reagent (Invitrogen), and 1 μg of RNA was reverse transcribed using a RevertAid First Strand cDNA Synthesis Kit (Thermo Scientific). The generated cDNA was subjected to real-time PCR with SYBR Green reagent (Roche, Basel, Switzerland). The primer sequences are listed in the Appendix A.

### 4.4. Clonal Sphere Formation Assay

Isolated single-cell suspensions of *Nes*-GFP^+^ cells were diluted to a density of 500 cells/mL, after which 2 µL/well of the diluted cell suspension was plated in ultralow attachment 96-well plates (Corning, Corning, NY, USA). Next, 150 µL of expansion medium was added to each well. Wells containing only one cell were marked and observed daily. After 10 days in culture, we recorded single cells that had generated spheres > 50 μm in diameter.

### 4.5. Cell Differentiation Ability In Vitro

Osteogenic differentiation: *Nes*-GFP^+^ cells were cultured in α-MEM (Invitrogen) containing 20% FBS (HyClone), 100 µg/mL ascorbic acid (Sigma-Aldrich), 100 nM dexamethasone (Sigma-Aldrich), 10 mM β-glycerophosphate (Sigma-Aldrich), and 100 IU/mL penicillin/streptomycin (Invitrogen) for 4 weeks. The medium was changed every 3 days. These osteogenic-differentiated cells were analyzed by staining with Alizarin Red to detect the presence of calcium.

Adipogenic differentiation: *Nes*-GFP^+^ cells were cultured in high-glucose DMEM containing 100 nM dexamethasone (Sigma-Aldrich), 10 µg/mL insulin (Sigma-Aldrich), 0.2 mM indomethacin (Sigma-Aldrich), 0.5 mM 3-isobutyl-1-methylxanthine (Sigma-Aldrich), 10% FBS (HyClone) and 100 IU/mL penicillin-streptomycin (Invitrogen) for 4 weeks. The medium was changed every 3 days. These adipogenic-differentiated cells were analyzed by staining with Oil red O.

Chondrogenic differentiation: *Nes*-GFP^+^ cells were cultured in a 15-mL conical tube with 2 mL of induction medium consisting of DMEM (Invitrogen) with 3% FBS (HyClone), 10 ng/mL tumor growth factor (TGF)-β3 (PeproTech), 1×ITS (Sigma-Aldrich), and 1 mM pyruvate (Sigma-Aldrich) for 4 weeks. The medium was changed every 3 days. Chondrocytes were identified by toluidine blue (Sigma-Aldrich) staining.

### 4.6. Flow Cytometric Analysis

T cells were stained for CD3-PB (Pacific Blue) and sorted with an Influx apparatus (BD), while flow cytometric analyses were performed with Influx or Gallios (Beckman Coulter, Brea, CA, USA) flow cytometers. Data were analyzed with FlowJo 10.6.2 software (Treestar, Woodburn, OR, USA). Additionally, we used counting beads (BD Biosciences, 340334, Franklin Lakes, NJ, USA) according to the manufacturer’s instructions to determine the % of *Nes*-GFP^+^ cells. The primary and secondary antibodies are listed in the Appendix A.

### 4.7. Immunofluorescence Staining

For immunofluorescence (IF) staining, the mouse spleens and transplants were fixed in 4% PFA and dehydrated with 30% sucrose. After fixation, the tissues were cut into 5-μm sections. The tissue sections were incubated with 5% normal serum and 0.1% Triton X-100 (HyClone) in PBS for 1 h at room temperature, followed by incubation with primary antibodies at 4 °C overnight. The sections were then incubated with secondary antibodies at room temperature for 1 h. Nuclei were counterstained with DAPI (4′,6-diamidino-2-phenylindole) for 10 min. The primary and secondary antibodies are listed in the Appendix A. Images were acquired using an LSM780 confocal microscope (Zeiss, Jena, Germany). For transplant staining, sections were obtained and stained with anti-mouse CD3-AF647 (green) or anti-mouse B220-AF594 (red) for 3 h at room temperature.

### 4.8. Artificial Lymphoid Formation

The protocol for the generation of artificial lymphoid-like structures was performed as follows: The isolated CD45^-^*Nes*-GFP^−^ and CD45^-^*Nes*-GFP^+^ cells derived from mouse spleens were placed onto a collagenous matrix (CS-35; Koken, Tokyo, Japan) and squeezed several times to allow cell adsorption into the scaffold. Scaffolds containing CD45^-^*Nes*-GFP^−^ cells were kept on ice and transplanted under one kidney capsule of anesthetized C57BL/6 mice, and CD45^-^*Nes*-GFP^+^ cells were transplanted under the other kidney capsule. After 3 weeks, we detected the lymphoid formation of these cells using specific antibodies against T cells (CD3) and B cells (B220). We collected the inguinal lymph nodes as lymph node controls.

### 4.9. Cytokine Assays

T cells were cultured with or without *Nes*-GFP^+^ cells for 2 days. During the last 6 h of incubation, PMA (50 ng/mL), ionomycin (500 ng/mL), and brefeldin A (BFA; 10 μg/mL) were added to the culture system (all from Sigma-Aldrich Aldrich). IFN-γ and TNF-α were analyzed by flow cytometry.

### 4.10. Proliferation Assay

Isolated CD3^+^ T cells were stained using a CellTrace Yellow cell proliferation kit (Invitrogen) according to the manufacturer’s instructions, and then were cultured with or without *Nes*-GFP^+^ cells treated with PHA (5 μg/mL) for 4 days. The percentage of T cell proliferation was detected by CFSE dilution.

### 4.11. T cells Activation

Isolated CD3^+^ T cells were cultured with or without *Nes*-GFP^+^ cells treated with PMA (100 ng/mL) for 24 h. The expression of CD69 and CD25 was analyzed by flow cytometry.

### 4.12. Inflammatory Bowel Disease (IBD) Model

Trinitrobenzenesulfonic acid (TNBS) (Sigma-Aldrich) was used to induce mice acute colitis. On day one, mice (C57BL/6, male, 6–8 weeks) were covered with 150 μL of a premixed solution containing TNBS after shaving back hair. On day seven, food was withdrawn from the mice for 24 h to remove as much stool as possible, after which we delivered TNBS (2.5 mg in 100 μL of 50% ethanol) into the colon 4 cm proximal to the anus of mice anesthetized with pentobarbital. The control mice received 100 μL of 50% ethanol. One day after the above treatment, the mice were subjected to intravenous (i.v.) injection of one million cells of spleen-derived *Nes*-GFP^+^ cells or bone marrow-derived *Nes*-GFP^+^ cells, respectively.

### 4.13. Contact Hypersensitivity Model

CHS reactions were induced in mice with 2,4-dinitro-1-fluorobenzene (DNFB) as previously described [38]. Briefly, DNFB (Sigma-Aldrich, 0.5% dissolved in 4:1 acetone/olive oil) was applied to the shaved back of each mouse (sensitization). After 5 days, the mice were challenged by epicutaneous application of 0.2% DNFB to the right ear. The control group consisted of mice challenged with 0.2% DNFB on the right ear without prior sensitization. Ear thickness was measured at 24, 48, 72, and 96 h post-challenge by individuals blinded to the treatment status. The degree of swelling was calculated as the thickness of the right ear (challenged ear) minus the baseline thickness of the left ear (unchallenged ear). To compare the immunosuppressive effects of *Nes*-GFP^+^ cells on the CHS response, the mice were subjected to intravenous (i.v.) injection of one million cells on day 2 post-challenge.

### 4.14. Statistical Analysis

All data are presented as the mean ± SD obtained from at least three independent experiments. Comparisons between the groups were performed using a one-way analysis of variance (ANOVA). Analysis was performed using SPSS (version 22). A *P* value less than 0.05 was considered statistically significant.

## Figures and Tables

**Figure 1 ijms-23-11819-f001:**
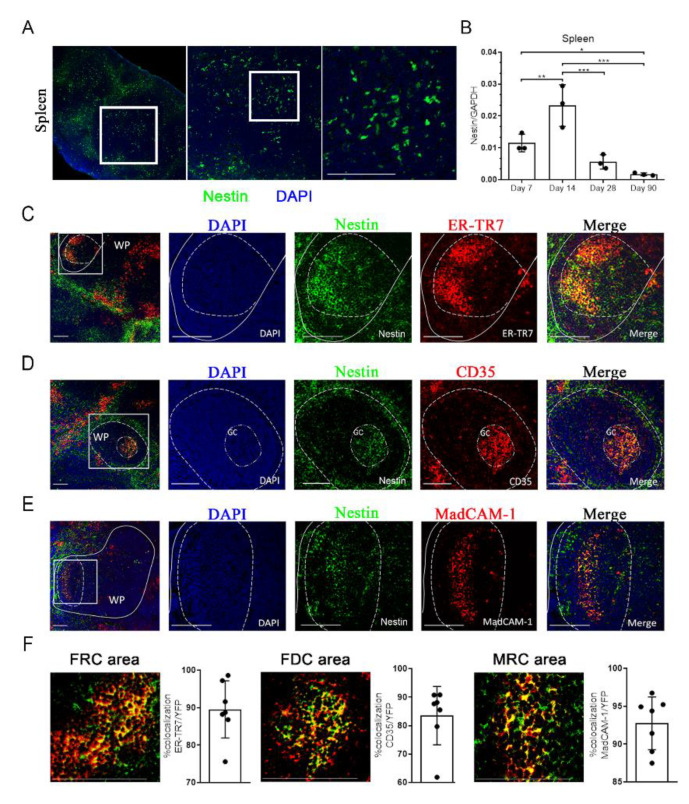
Characterization of Nestin expression in the spleen: (**A**) Nestin (green) expression in the spleens of *Nes*-GFP transgenic mice. Nuclei were counterstained with DAPI (blue). Scale bar, 100 µm; (**B**) qRT-PCR analysis of Nestin gene expression in the spleen of normal mice on days 7, 14, 28, 90 after birth. The relative mRNA expression levels of Nestin were normalized to reference GAPDH controls. *n* = 3. (**C**–**E**) Immunohistochemical analyses showed that the *Nes*-GFP^+^ cells in the spleen expressed markers of FRCs, FDCs, and MRCs; (**C**) ER-TR7 (FRCs); (**D**) CD35 (FDCs); (**E**) MadCAM-1 (MRCs). Nuclei were counterstained with DAPI (blue). Scale bar, 50 µm. White Pulp, WP; Germinal Center, GC; (**F**) The co-expression area of *Nes*-GFP^+^ cells with ER-TR7, CD35, and MadCAM-1 was counted. *n* = 7.Data represent mean values ± SD of three independent experiments. * *p* < 0.05, ** *p* < 0.01, *** *p* < 0.001.

**Figure 2 ijms-23-11819-f002:**
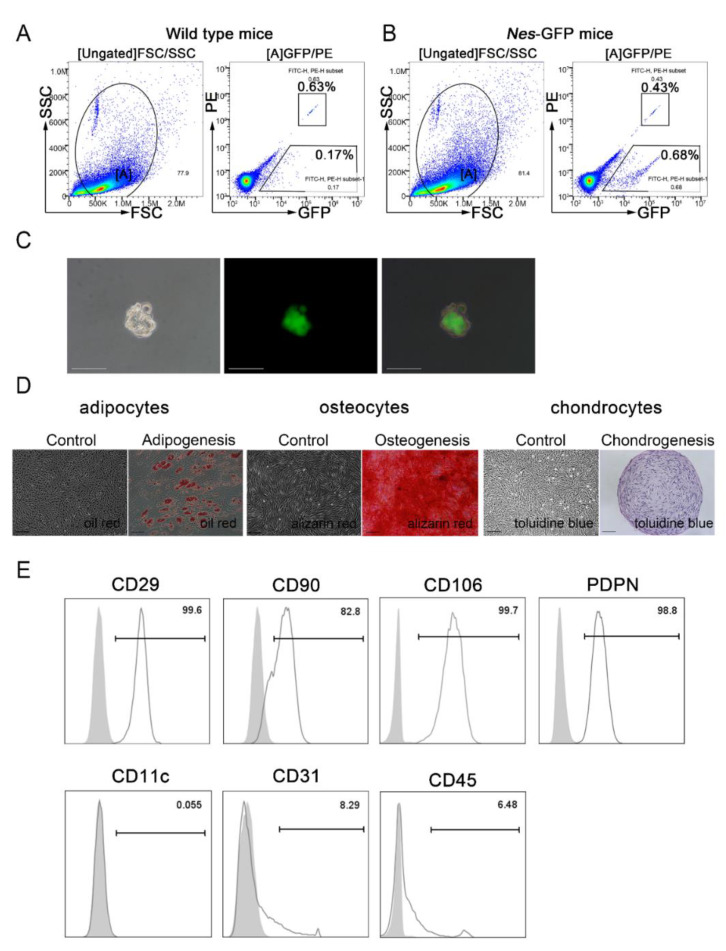
Isolation and characterization of *Nes*-GFP^+^ cells from the spleen. (**A**,**B**) Flow cytometry was used to isolate *Nes*-GFP^+^ cells from the spleens of *Nes*-GFP transgenic mice. Cells from wild-type C57BL/6 mice were isolated as a control. To detect the percentage and number of *Nes*-GFP^+^ cells in cell sorting more precisely, we added the counting beads (both positive in PE and GFP channel) in the cell suspension of the spleen. (**C**) Representative images showing the clonal sphere growth of single *Nes*-GFP^+^ cells were observed using a bright field (left panel) or fluorescence (middle panel). Scale bars, 50 μm. (**D**) Representative stained images showing that mouse spleen-derived *Nes*-GFP^+^ cells could differentiate into osteocytes (Alizarin red), adipocytes (Oil red O), and chondrocytes (toluidine blue). Scale bars, 100 μm. (**E**) The expression of cell surface markers on *Nes*-GFP^+^ cells was detected by flow cytometry.

**Figure 3 ijms-23-11819-f003:**
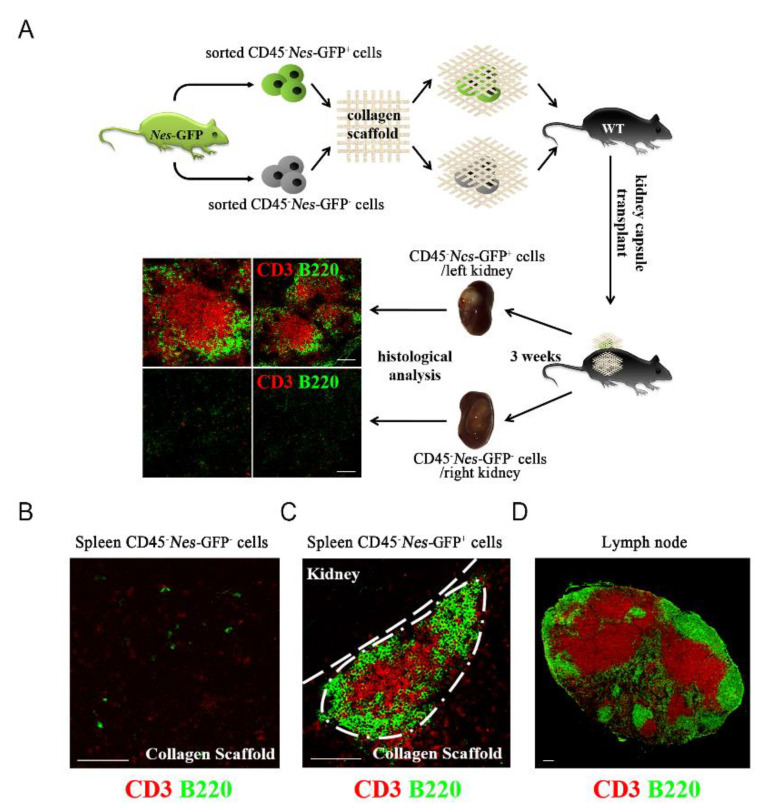
Spleen-derived *Nes*-GFP^+^ cells could form secondary lymphoid organs. (**A**) Spleen CD45^-^*Nes*-GFP^−^ and CD45^-^*Nes*-GFP^+^ cells of *Nes*-GFP transgenic mice with collagen scaffolds were transplanted into the kidney capsule of wild-type mice. Scale bars, 100 μm. (**B**,**C**) Representative stained images of transplants showed that *Nes*-GFP^+^ cells, instead of *Nes*-GFP^-^ cells, could form lymphoid tissue-like organoids. Scale bars, 100 μm. (**D**) Representative stained images of lymph node. Scale bars, 100 μm.

**Figure 4 ijms-23-11819-f004:**
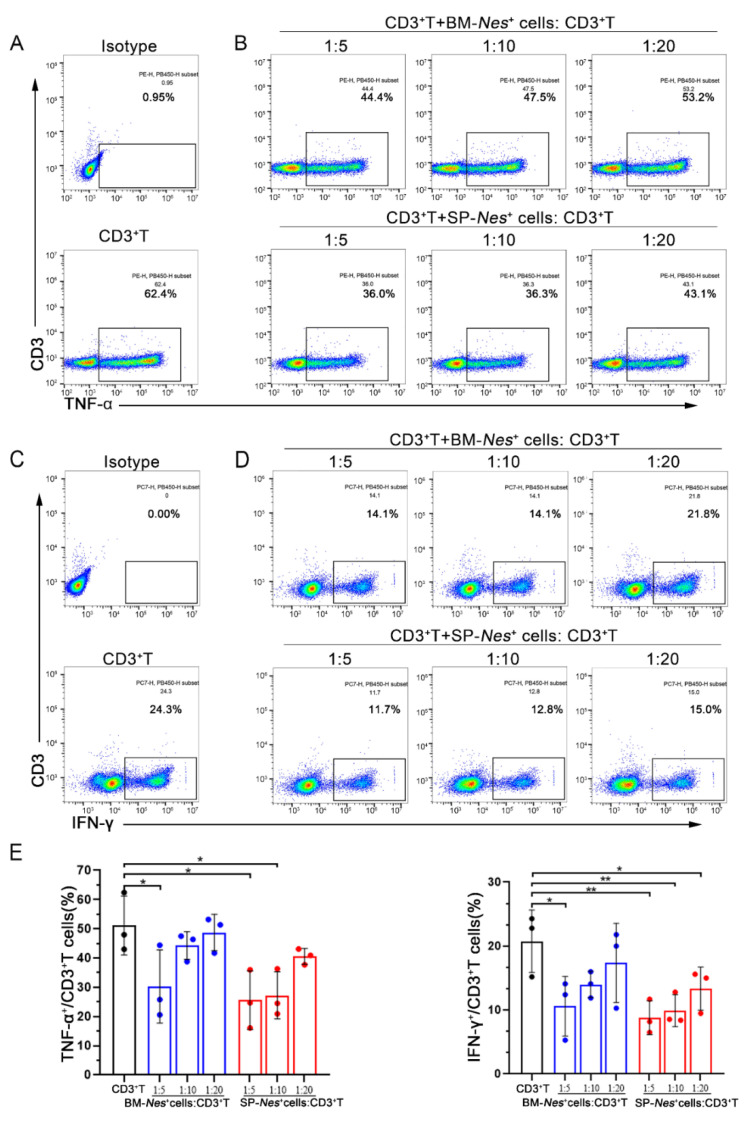
Immunosuppressive properties of *Nes*-GFP^+^ cells derived from the spleen (cytokine assay). (**A**,**B**) *Nes*-GFP^+^ cells were cocultured with sorted CD3^+^ T cells, and the pro-inflammatory cytokine TNF-α secretion of T cells was evaluated by flow cytometry. Representative plots of TNF-α production by T cells at different ratios. *n* = 3. (**C**,**D**) *Nes*-GFP^+^ cells were cocultured with sorted CD3^+^ T cells, and the pro-inflammatory cytokine IFN-γ secretion of T cells was evaluated by flow cytometry. Representative plots of IFN-γ production by T cells at different ratios. *n* = 3. (**E**) Bar graphs showed the inhibition rate of IFN-γ- and TNF-α-producing CD3^+^ T cells after coculture with *Nes*-GFP^+^ cells. Data represent mean values ± SD of three independent experiments. * *p* < 0.05, ** *p* < 0.01.

**Figure 5 ijms-23-11819-f005:**
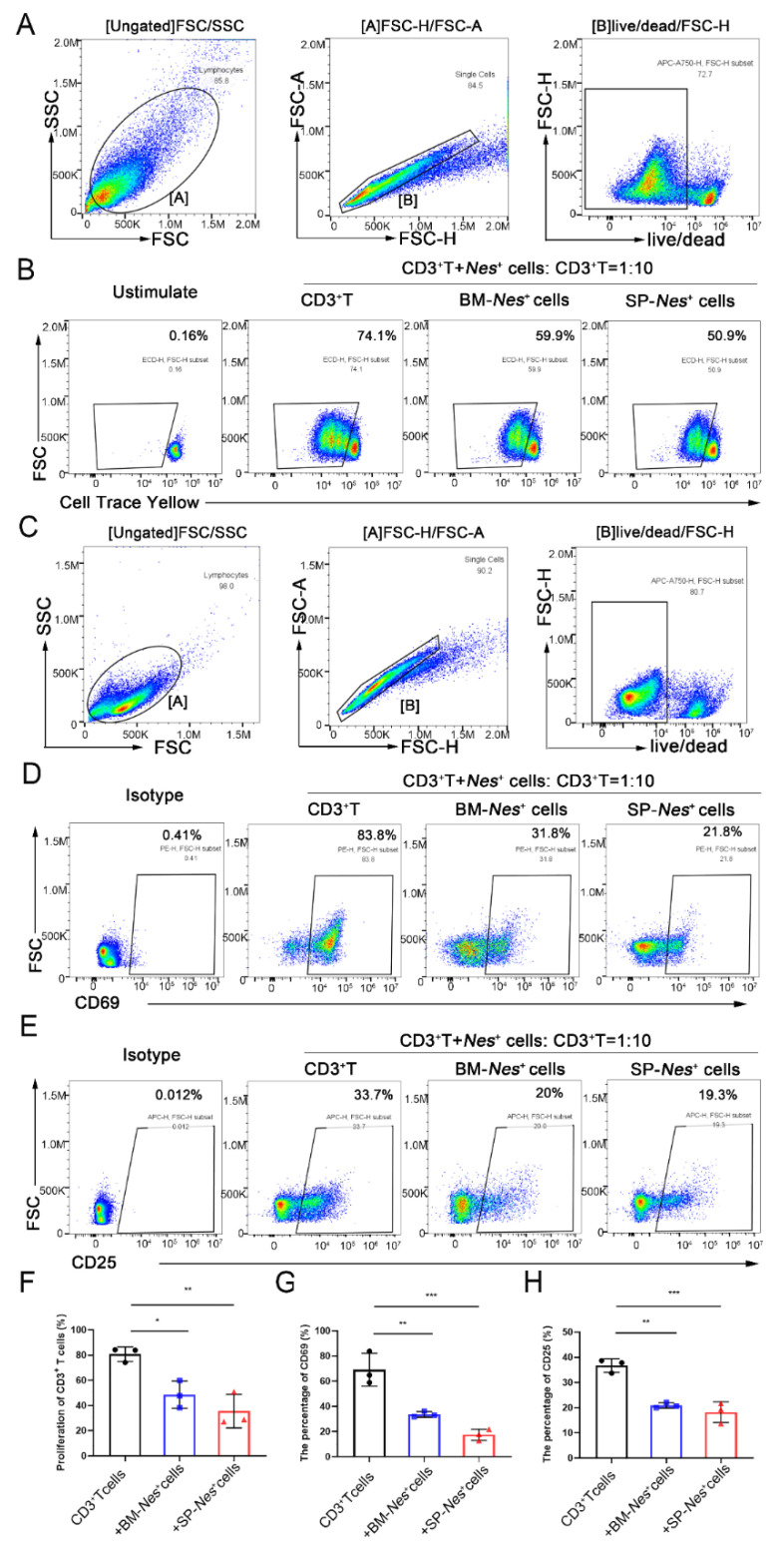
Immunosuppressive properties of *Nes*-GFP^+^ cells (T cells proliferation and activation suppression assay): (**A**) Gating strategy for the evaluation of T cells proliferation; (**B**) *Nes*-GFP^+^ cells were cocultured with sorted CD3^+^ T cells, and the percentage of T cells proliferation was evaluated by flow cytometry. Representative plots of three independent experiments. (**C**) Gating strategy for the evaluation of T cells activation markers (CD69 and CD25); (**D**,**E**) *Nes*-GFP^+^ cells were cocultured with sorted CD3^+^ T cells, and the percentage of CD69 and CD25 was evaluated by flow cytometry. Representative plots of three independent experiments; (**F**–**H**) Bar graphs showed the inhibition rate of T cells proliferation and activation after coculture with *Nes*-GFP^+^ cells. Data represent mean values ± SD of three independent experiments. * *p* < 0.05, ** *p* < 0.01, *** *p* < 0.001.

**Figure 6 ijms-23-11819-f006:**
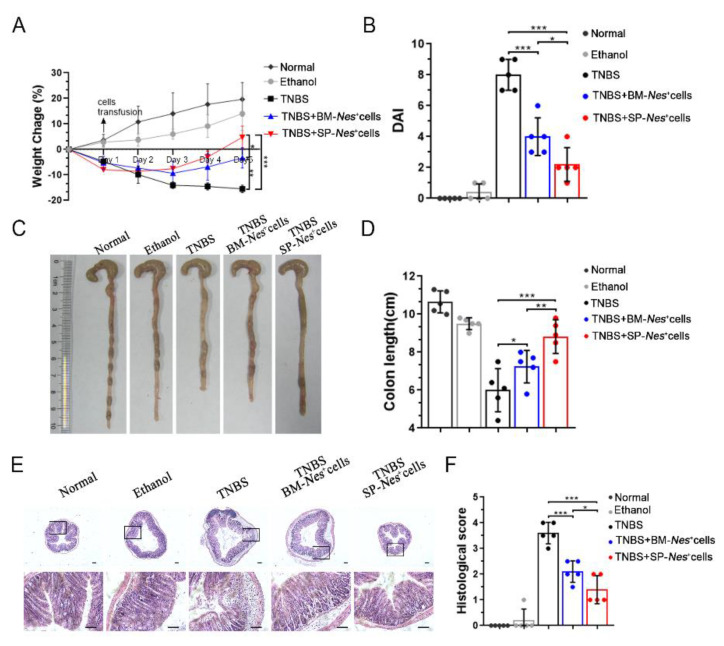
Spleen-derived *Nes*-GFP^+^ cells as a potential source for the treatment of inflammatory diseases: (**A**) Therapeutic efficacy evaluation of TNBS-induced IBD model, bodyweight change percentage of each group. *n* = 3; (**B**) Disease activity index (DAI) were measured. *n* = 5; (**C**) Representative colonic length of mice; (**D**) Quantification of the colonic length of mice. *n* = 5; (**E**,**F**) H&E staining and corresponding histological scoring of each group. *n* = 5. Scale bars, 100 μm. * *p* < 0.05, ** *p* < 0.01, *** *p* < 0.001.

## Data Availability

All relevant data and materials are available from the authors upon reasonable request.

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
