# Peer review of "Nestin+ Mesenchymal Precursors Generate Distinct Spleen Stromal Cell Subsets and Have Immunomodulatory Function"

_ijms, 2022, doi:10.3390/ijms231911819_

Round 1

Reviewer 1 Report

In this study, Huang J et al sort splenic nestin+ cells and they characterize the features of this cell population. The authors prove that splenic nestin+ cells have mesenchymal stromal cell (MSC) properties and they exert a larger immunosuppressing function compared to MSCs from the bone marrow. The authors used a wide variety of methods, such as immunostaining, in vivo experiments, cell sorting, flow cytometry etc. Since MSCs are considered as a clinically important cell type, the scientific question of this manuscript may be of potential interest. The manuscript is in general logical, easy to follow. However, the nestin positivity of splenic cells has already been described (Avanzini MA et al, Am J Hematol. 2018 May;93(5):615-622) and a careful comparison of nestin+ and nestin negative cells lacks that could be an important point of the work.

Major comments:

1. An interesting point would be to characterize the heterogeneity of Nes-GFP+ cells that could increase the novelty of the work. Nes-GFP+ cells show a heterogeneity for CD31 and CD45 with 6-8% being positive for these markers. Could the authors sort these cells and carry out some further studies, such as RT-qPCR analysis?

2. Fig. 2 and fig. 4: Do nestin negative cells behave similarly to nestin+ cells? Do nestin negative cells have an immunosuppressing effect, clonal sphere forming and differentiating capability? These are critical questions to show that nestin+ cells have different features compared to nestin negative cells.

3. Figure 4B, E: in contrast to the manuscript text (line 162), I do not see a significant dose-dependent effect for SP-Nes+ cells and TNFa or IFNg production. How can the authors explain that irrespective of the co-culturing ratio, there is no change in the magnitude of the effect?

Minor comments:

4. The manuscript text should not contain all the exact statistical values.

5. Figure 2A, B: why is PE signal shown on the dot plots (y-axis)?

6. Figure 4B, E: although there seems to be some dose-dependent effect for BM-Nes+ cells and TNFa, this is not obvious on Fig 4B. How can the authors explain this discrepancy?

7. Figure S1A: please show control images (e.g. from R26RtdT mice) to prove the specificity of the mouse model.

8. Figure S1B: overlapping among the markers is not convincing, especially for VCAM-1. Could you please show images with higher magnification?

9. Figure S2B, C: I think these figures should contain CD3+CD8+ T cells in their right panels.

1

Reviewer 2 Report

In general I can see a good manuscript. Please assume responsability about the novelty of describing this cell population or cite previous references and highlight the comparison role of the manuscript versus Bone marrow MSCs. 

I have some comments as well as requests to make the manuscript suitable for being published and in the attempt to improve it. 

1. The organization of the manuscript is a bit unusual, the choice to have all figures in a separate section should be discussed with assistant editor. However, I kindly suggest in advance to make modification, as recommended in IJMS instructions for authors, that are "You must insert your graphics (schemes, figures, etc.) in the main text after the paragraph in which it is first cited."

2.. When the authors said in the manuscript introduction section (line 44th) "...therapeutic potential in tissue injury repair and inflammatory disease", they can add details about such potential, such as including and mentioning the antimicrobial properties of MSCs (ref. doi.org/10.3390/antibiotics10070750), for example, properties extremely connected to the immunomodulatory function, indeed.

3. the use of CD45-Nes-GFP+ with the symbol " - " that means negative expression in comparison then to other abbreviation such as Nes-GFP+, that in this other case means co-expression with GFP, is confusing. 

Please, I highly suggest to use superscript for positive or negative expression of any called marker during cell specification in the main text and especially in figures. The symbol " - " should then used only as conjunction, such as for Nes-GFP mice or Nestin-fluorescent cells.  

4. In figure 2 panel A and B, please remind to the reader in the legend PE fluorescence meaning.  

5. Negative controls, I mean not induced samples, should be added besides the trilineage differentiation capacity shown in fig 2D. 

6. Details about the source of Lymph node controls shown in panel 3D should be added to supplementary informations. Is from wt mice or GFP mice? same animal subjected to transplant or not? This info will be interesting, thanks

5. From what I can read in the results and associated legend, Figure 4 show sorted T-cell ratio and their analysis when immunosuppressive activity was evaluated in co-culture with spleen Nes-cells. The axis shown in Fig 4, panels A,B,C and D, are tricky and the scale is also in contrast between the x-axis and y-axis. All cells should be CD3+ (positive)(if T c-cells are represented, right?), however the scale starting from 10^2 seems to do not highlight this and likely makes confusion, thinking accidentally that maybe spleen cells are shown in the isotype and below plot, or that is a missing marker in sorted known CD3+ (positive) cells. On the contrary the signal of TNF-a and IFN-y is clear. Please amend this graphical, but significant issue.  

6. I believe the data in Supplementary figure 3, which shows the T-cell activation profile, deserves to be included in the main figure manuscript and not in supplementary. That's because general future similar results will have to be compared in other researchers' future works.
I respect instead the choice to provide the fig S2 as supplementary info.

7. More a comment than a request. Figure S4 is OK in supplementary since not directly related to IBD, but only to specific effect in vivo of immunomodulatory cells, however, as the authors declared this is the first characterization of the focused cell type, this results may be moved to the main manuscript. Your choice other referees opinion. 

8.Please check that material and methods that were used exlusively for supplementary figures and supplementary results have to or not included in the main manuscript following MDPI editorial rules.

9. Do you have any preliminary info about B-cells number and this spleen Nestin+ Mesenchymal Precursors ? this is a scientific curiosity and maybe a new brief discussion point for the authors concerning the resting immunomodulatory function.

Round 2

Reviewer 1 Report

Attached Figure 1 and Figure 2 may be interesting not only for the reviewer, but for the readers of the manuscript, too. Thus, attached Fig1 and 2 should be included into the paper as normal figures or as supplementary figures.

Author Response

We fully agree with the reviewer's comment. As suggested, we added the Attached Figure 1 and Figure 2 into our supplementary information as figure S2. 

Reviewer 2 Report

All my comments were sufficiently, properly and kindly addressed. 

Author Response

Thanks for the reviewer's comment.